# Back Pain Prevalence, Intensity, and Associated Risk Factors among Female Teachers in Slovakia during the COVID-19 Pandemic: A Cross-Sectional Study

**DOI:** 10.3390/healthcare9070860

**Published:** 2021-07-07

**Authors:** Pablo Prieto-González, Miroslava Šutvajová, Anna Lesňáková, Peter Bartík, Kristína Buľáková, Teresa Friediger

**Affiliations:** 1Health and Physical Education Department, Prince Sultan University, Riyadh 11586, Saudi Arabia; pprieto@psu.edu.sa (P.P.-G.); pbartik@psu.edu.sa (P.B.); 2Faculty of Health, Catholic University, 03401 Ružomberok, Slovakia; sutvajova.m@gmail.com (M.Š.); kristina.bulakova@gmail.com (K.B.); t_grzeda@o2.pl (T.F.); 3Hospital with Polyclinic MUDr. L. Nádaši Jégého, 02601 Dolný Kubín, Slovakia; 4Central Military Hospital SNU-FH Ružomberok, 03401 Ružomberok, Slovakia

**Keywords:** back pain, female teachers, COVID-19 pandemic, home office

## Abstract

Significant back pain prevalence and intensity are reported in female pedagogues. Due to the COVID-19 pandemic, they have been exposed to remote working conditions, higher psychological demands, and stress. Our objective was to evaluate the back pain prevalence, intensity, and related risk factors in female teachers from Slovak regions in the context of remote learning during the COVID-19 pandemic. An electronic questionnaire was applied for data collection. A total of 782 adult female teachers (age 43.11 ± 0.36, BMI 34.94 ± 5.94) from primary, secondary, and tertiary schools from Slovak regions were included in the survey. Of these, 74% reported cervical pain, 67% lower back pain, and 60% pain in more than one vertebral region. The highest pain intensities were reported in the following cohorts (pain scale 1–5, 1 = no pain, and 5 = extreme pain): pedagogues from the Presov region (3.74), those working in the special education sector (3.83), those who gave five online classes a week (3.58), those who performed no physical activity (PA, 3.86), pedagogues who did not work in compliance with ergonomic recommendations (3.59), those with moderate or severe stress (3.66), and those who were sitting all or most of the time (3.62). The main risk factors identified were: no PA performed, no compliance with ergonomic recommendations, and stress.

## 1. Introduction

Back pain is a widespread and common problem that most people experience at some point in their lives [1]. It affects mostly adults, causes disability for some, and is the most common reason for seeking healthcare [2,3]. It is a serious cause of work absenteeism and related expenses. This serious musculoskeletal disorder (MSD) affects the quality of life in pedagogues, resulting in frequent sick leave, functional impairment, absenteeism, and early retirement [4]. Pedagogues are exposed to a considerable risk of this pain arising [5]. Teachers at schools included in studies reported a high prevalence of MSD, ranging anywhere between 39% and 95% [6,7,8,9]. Worldwide, back pain is one of the most common musculoskeletal disorders, ranking among the top 10 most frequent health issues [10]. The point prevalence of lower back pain in teachers is reported as follows: 21.8% [11], 38.1% [12], 40.4% [4], 45.6% [13], 64.98% [14], and 74.8% [15], and the point prevalence of cervical spine pain has been reported as: 11.3% [12], 24% [16], 48.7% [13], and 69.3% [17]. Wáng et al., (2016) reported higher back pain prevalence in females than in males, especially after menopausal age, in their systematic review [18]. In addition, female teachers report more depressive symptoms than males, which are considered significant risk factors for MSDs [19]. 

In Europe, where the female gender dominates, the proportion of female teachers has reached 90% in 11 member states, with the highest totals in Lithuania, Hungary, Slovenia (97% in each), and Italy (96%). Thus, 85% of teachers working in the EU are women [20]. Therefore, the educational sector may be vulnerable to severe work absenteeism and performance reductions in the future based on the predisposition of women to back pain and MSD, in addition to the circumstances of the ongoing COVID-19 pandemic. 

According to WHO (2019), quarantine and personal protection are two of the best ways to limit infectious respiratory diseases. Therefore, various measures were implemented worldwide to prevent the spread of the new coronavirus [21]. In the case of Slovakia, between 18 December 2020 and 8 February 2021 a state of emergency was declared. Thus, parallel to the deteriorating epidemiological situation and the curfew enforcement by the authorities, all departmental institutions and schools were closed. Furthermore, some restrictive measures implied the limitation or suspension of fundamental rights, such as the right of free movement and the right of assembly. Similarly, all public cultural and sporting activities were suspended [22]. These measures may have negatively impacted individual physical activity (PA) in terms of intensity and frequency, dramatically increasing emotional stress levels [23]. In this regard, it must be underlined that a sedentary lifestyle and low PA levels have long-term adverse effects on the health, well-being, and quality of life of individuals. 

Furthermore, the home office environment predisposes individuals to inappropriate positions that can cause pain and changes in the musculoskeletal system, especially in the spinal region [24]. Considering the abovementioned facts, investigating the current situation and verifying the back pain prevalence in female pedagogues is highly important for mapping actual problem development. Therefore, the main objective of the research was to analyze the back pain prevalence, intensity, and related risk factors in female pedagogues in the context of remote learning during the COVID-19 pandemic. We noted that no other studies reported back pain prevalence in Slovak pedagogues before or during the pandemic. The study was conducted during the application of strict measures between 18 December 2020 and 8 February 2021 in Slovakia. In this context, we hypothesized that the situation of MSD prevalence and pain intensity would worsen. Additionally, the obtained data are relevant not only for the pandemic situation but also for the future with regard to the growing sector of remote learning.

## 2. Methods

### 2.1. Ethical Statement

The study was conducted in compliance with the principles set out in the Helsinki Declaration (revised in 2013) [25] and the guidelines of the Council for International Organizations of Medical Sciences (CIOMS), in compliance with the International Ethical Guidelines for Health-Related Research Involving Humans (2016) [26], and in compliance with the Guidelines for Biomedical and Health Research Involving Human Participants (2017) [27]. The purpose of the survey and its essential aspects (objective, extension, and description) was detailed within the open electronic online survey framework. By submitting the completed questionnaire, respondents granted their permission to have it processed. No additional statement of permission was required as part of the survey questionnaire. 

### 2.2. Subjects and Survey Development

A total of 782 female teachers (age 43.11 ± 0.36, BMI 34.94 ± 5.94) voluntarily participated in the survey. The research was conducted by the Department of Physiotherapy at the Faculty of Health of the Catholic University in Ruzomberok, Slovakia. A self-administered questionnaire composed of 18 questions was used to conduct this cross-sectional study. As per the inclusion criteria for the study, the participants were: (1) Pedagogues in Slovakia during the introduction of online classes during the COVID-19 pandemic in January 2021; (2) who were working in Slovakia in primary, secondary, tertiary, or special needs schools; and (3) who were aged between 18 and 65 years. As per the exclusion criteria, individuals who were: (1) of male gender; or (2) working in kindergartens were excluded. The survey contained 3 parts. The first part included a series of demographic aspects. The second contained information related to the practice of physical activity during COVID-19 lockdown. The third part included items related to work characteristics, musculoskeletal pain intensity, functional impairment, psychological-emotional exhaustion, and LBP ergonomic recommendations. Demographic questions included in the survey concerning age, weight, height, gender, academic titles, regions of activity, number of years in the field, and the type of school respondents taught. The questionnaire was compiled using the online Google Forms.

Once the target population of the study was defined, a simple random selection method via social media such as Facebook, WhatsApp, and Messenger was used. The questionnaires were distributed on social media on 15 January 2021 and collected on 22 January 2021. Men were excluded from the survey due to the inequality of representation within the responses collected. 

### 2.3. Questionnaire Validation

All items were written in simple, short, and plain language to enable a better understanding of the questions. The survey responses were structured on a scale from 1 to 5, where 1 refers to non-adherence to statement and 5 to 100% adherence. Similarly, pain intensity was rated from 1 to 5, with 1 being the absence of pain and 5 being extreme pain. Five experts were chosen to clarify the quality and reliability of the questions presented. Additionally, the reliability was verified by the Cronbach’s alpha validation formula. For this purpose, a pre-trial with 30 subjects completing the survey was conducted. The Cronbach’s alpha was calculated with a value of 0.89, indicating high internal consistency. Moreover, the present study adheres to STROBE guidelines [28]. 

### 2.4. Sampling Process

The required sample size for this study was calculated using the following formula:n = Z2p × qN/e2 (N − 1) + Z2p × q

where n = sample size, N = population size, Z = confidence level, p = probability of success, q = probability of failure, and e = confidence interval. The confidence level was set at 99%, the confidence interval at 5%, and the probability of success at 50%. The estimated population of female teachers in Slovakia is 56,000. Therefore, once the calculation was performed, it was verified that the sample size required to represent the studied population was 658 subjects. The incomplete submission of the questionnaire was precluded by the Google forms’ functionality which does not allow the submission of partially answered or incomplete questions, thus avoiding the need for an increased sample size. The data were collected using a self-administered structured questionnaire distributed by Google Forms. Thus, once the number of responses reached 782, the open online link for this e-survey was closed so that it could not accept more responses and the analysis could be performed.

### 2.5. Survey Management

The cross-sectional online survey was sent to teachers during the period of introduction of online distance education during the COVID-19 lockdown in January 2021. 

Potential participants were addressed through online forums for teachers in Facebook groups where the link to the electronic questionnaire was published. Submission of the completed questionnaire was considered synonymous with permission to participate in the survey. The social media groups WhatsApp and Facebook - Messenger were used to disseminate the questionnaire. The electronic survey is cost-effective, time-effective, and ecological, and during online distance teaching when in COVID-19 lockdown, it was a practical solution for obtaining necessary data. The data analysis was performed using the SPSS program.

### 2.6. Statistical Analysis

The data normality was assessed using the Kolmogorov–Smirnov test, whereas homoscedasticity was verified using Levene’s tests. To establish comparations between 2 samples, Student’s t-test was used. To set comparisons between more than 2 sets of data, ANOVA with Tukey´s post hoc test was conducted. However, when the homogeneity of variance was violated, comparisons between 2 cohorts were made using the Mann–Whitney U test, whereas the Kruskal–Wallis H test was conducted to establish comparisons between more than 2 groups, applying the Dunn–Bonferroni post hoc test for pairwise comparisons. To estimate the effect size (ES), the η2 parameter was used when the comparisons were made with parametric tests. For non-parametric comparations, after performing the Mann–Whitney U test, the following formula was used: ES = Z-score/√Number of observations. An ES of 0.2 was considered small, 0.5 moderate, and 0.8 large. To analyze the associations between dependent and independent variables, the Pearson correlation coefficient was used, and the results were interpreted as follows: r = 0 null correlation; 0.01 ≤ r ≤ 0.09 very weak; 0.10 ≤ r ≤ 0.29 weak; 0.30 ≤ r ≤ 0.49 moderate; 0.50 ≤ r ≤ 0.69 strong; and r ≤ 0.70 very strong [29]. The significance level was set at 0.05. Data are presented as mean (SD). All statistical analysis was performed using SPSS version 26 (Chicago, IL, USA).

## 3. Results 

As shown in Table 1, almost three-quarters of survey respondents reported suffering from cervical pain, and more than two-thirds from lower back pain. Likewise, more than 60% of the interviewees reported suffering from pain in more than one vertebral region. The percentage of subjects who declared suffering from pain in the thoracic spine was slightly less than 30%. Only 5.36% of the respondents stated that they did not suffer from vertebral pain.

Table 2 shows the comparisons established between different cohorts, subgroups, or conditions. The Kruskal–Wallis H test determined the existence of a main effect of the region variable [(*p* < 0.001); df (7, 774)]. Then, the Dunn–Bonferroni post hoc test revealed that the respondents from Trnava presented a pain intensity that was significantly lower as compared to Banska Bystrica interviewees (*p* < 0.001; ES: 0.41) and those from Presov (*p* = 0.014; ES: 0.49). Nitra interviewees reported significantly less pain than Banska Bystrica respondents (*p* <.001; ES: 0.27). Zilina interviewees reported significantly lower spinal pain intensity as compared to those from Bratislava (*p* = 0.006; ES: 0.13), Banska Bystrica (*p* < 0.001; ES: 0.17), Kosice (*p* = 0.0; ES: 0.29), and Presov (*p* < 0.001; ES: 0.34). Respondents from Bratislava reported suffering from significantly lower spinal pain intensity as compared to those from Banska Bystrica (*p* = 0.001; ES: 0.19). Banska Bystrica survey respondents stated they suffered significantly lower spinal pain intensity as compared to those from Trencin (*p* < 0.001; ES: 0.15), Kosice (*p* < 0.001; ES: 0.21), and Presov (*p* = 0.001; ES: 0.27). 

A main significant effect was observed for the school type factor [(F = 3, 778); (*p* = 0.004)]. The pairwise comparison showed that the spinal pain reported by teachers from secondary schools was significantly higher than the pain reported by primary school teachers. 

There was also a significant main effect for the variable number of days a week of online classes ((F = 4, 777); (*p* < 0.001)). The pairwise comparisons showed that the spinal pain intensity reported by the respondents who taught online classes once a week was significantly higher than in those who taught online classes twice a week (*p* < 0.001; ES = 0.18), but significantly lower than the pain reported by the interviewees who taught online classes thrice a week (*p* < 0.001; ES = 0.15), four times a week (*p* < 0.001; ES: 0.16), and five times a week (*p* < 0.001; ES= 0.14). The spinal pain reported by respondents who taught online classes twice a week was significantly lower as compared to those who taught thrice a week (*p* < 0.001; ES = 0.12), four times a week (*p* < 0.001; ES = 0.15), and five times a week (*p* < 0.001; ES = 0.19). Finally, the interviewees who taught online classes twice a week declared suffering from a significantly lower spinal pain intensity as compared to those subjects who taught four times a week (*p* < 0.001; ES = 0.11), and five times a week (*p* < 0.001; ES = 0.17).

A main significant effect was also observed for the variable *weekly practice of PA* during quarantine ((F = 4, 477); (*p* < 0.001)). The post hoc test revealed that the spinal pain intensity of the subjects who did not practice PA was significantly higher than in those who practiced PA three or four times a week (*p* = 0.008; ES = 0.17) and those who practiced PA five or six times a week (*p* < 0.001; ES = 0.67). Similarly, the spinal pain reported by the subjects who practiced PA once a week was significantly higher than reported in those who practiced PA five or six times a week (*p* = 0.024; ES = 0.51).

Significant differences were found between the spinal pain intensity reported by the subjects who complied with the ergonomic recommendations and those who did not (*p* < 0.001; ES = 0.17). Similarly, the interviewees who reported suffering mild or no stress reported significantly lower back pain as compared to individuals who suffered from moderate to severe stress (*p* < 0.001; ES = 2.04). Finally, the spinal pain intensity reported by the respondents who were sitting all or most of the time was significantly higher than those who were moving all or most of the time (*p* < 0.001; ES = 1.87).

No significant differences between cohorts were found for the following three factors: BMI, years of teaching, and weekly teaching hours.

The association between spinal pain intensity and personal and environmental variables is shown in Table 3. A significant negative correlation was found between spinal pain intensity and two factors: weekly practice of PA and compliance with ergonomic recommendations. Besides, a significant positive correlation was observed between pain intensity and perceived stress. However, no significant correlation was found between pain intensity and the following factors: age, BMI, years of teaching, number of days a week having online classes, weekly teaching hours, and time spent sitting or moving.

## 4. Discussion

The present cross-sectional study was conducted to: (a) examine back pain prevalence and intensity among Slovak female teachers in the context of remote learning during the COVID-19 pandemic; and (b) assess the effect of different risk factors on back pain intensity.

The prevalence of musculoskeletal disorders increases with age. However, those disorders are becoming relatively common in younger adults [30]. This phenomenon is probably related to female teachers’ average age, which is currently 43.11 ± 0.36, and is close to the EU average. In 2014, one out of every three teachers at EU primary schools was 50 years of age or older. In Italy, more than half the teachers belonged to this age group (53%). Similar figures were also found in Lithuania (50%), Estonia (49%), and Bulgaria (48%) [20].

In the present study, the results revealed that the highest prevalence of musculoskeletal symptoms in the period between December 2020 and January 2021 occurred in the cervical spine (74.84%), followed by the lumbar spine (67.68%), and the thoracic spine (29.12%). Previous studies have found that the prevalence of musculoskeletal symptoms and low back pain is associated with sitting and working with a computer or laptop in individuals with sedentary jobs [31,32]. The prevalence ranges from 40% to 80% [33].

Comparisons showed that the intensity of back pain reported by respondents who taught online courses once a week was significantly higher than the intensity in those who taught online courses twice a week (*p* < 0.001; ES = 0.18). However, this intensity was significantly lower compared to those who taught online three times a week (*p* < 0.001; ES = 0.15), four times a week (*p* < 0.001; ES: 0.16), or five times a week (*p* < 0.001; ES = 0.14). The pain intensity reported by those who taught online twice a week was significantly lower than that reported by those who taught online three times a week (*p* < 0.001; ES = 0.12), four times a week (*p* < 0.001; ES = 0.15), or five times a week (*p* < 0.01; ES = 0.19). Finally, respondents who taught online twice a week declared that their back pain intensity was significantly lower as compared with those subjects who taught four times a week (*p* < 0.001; ES = 0.11) or five times a week (*p* < 0.001; ES = 0.17). According to Daneshmandi et al., (2017), the persistence and increase in pain intensity of these problems can be attributed to static positions [33].

The results also showed that the severity of musculoskeletal pain in different body areas was related to different aspects such as PA and stress. Higher pain intensity was associate with the following risk factors: living in the region of Presov (3.74), working in the special education sector (3.83), teaching online classes five days a week (3.58), 20–29.9 years of teaching (3.66), 30–39.9 h of classes per week (3.8), not practicing PA (3.86), not complying with ergonomic recommendations (3.59), moderate or severe levels of stress (3.66), and sitting all or most of the time (3.62). More than 60% of the interviewees indicated that they suffered from back pain in more than one spinal region. The percentage of respondents who reported pain in the thoracic spine was just below 30%. Only 5.36% of participants declared no back pain. MSD studies carried out with teachers confirm this finding. The lower back is the most commonly affected anatomic region for pain [4,5,34,35]. A sedentary lifestyle at a young age may lead to chronic diseases in adulthood [36]. The European Agency for Safety and Health at Work states that musculoskeletal disorders in Europe represent the main cause of work absenteeism [37]. Several studies have found a high prevalence of musculoskeletal pain among teachers. In high schools, in the region of Aljouf (Saudi Arabia), Abdel-Salam (2019) reported that 68.5% of teachers suffered from musculoskeletal pain in different body areas [5]. In the present study, it was also found that secondary school teachers presented significantly more spinal pain than primary school teachers. In this regard, the studies conducted by Thaseen and Tantry (2019) and Mohseni-Bandpei et al., (2014) highlighted the greater probability of suffering from LBP in secondary school teachers as compared to those teaching in primary schools [11,38]. The reasons are factors such as age, body mass index, working years, work satisfaction, and work activities [38]. The occupation of teaching is both physically and mentally demanding in secondary schools. Consequently, there was a high incidence of cervical and lumbar pain during the online teaching period.

Our results showed that teachers who performed PA three or four times a week during the pandemic (*p* = 0.008; ES = 0.17) had a significantly lower back pain intensity than those who did not. Similarly, the prevalence of back pain reported by the teachers who practiced PA once a week was significantly higher than those who exercised five or six times a week (*p* = 0.024; ES = 0.51), and those who did not exercise (*p* = 0.008; ES = 0.17). These results coincide with the study conducted by Alzahrani et al., (2019) [39]. They found that PA was associated with a lower prevalence of LBP [39]. Furthermore, it was also found that practicing PA six or seven times a week could reduce back pain during the lockdown. In contrast, and based on the present study results, the absence of PA practice could generate the opposite effect. At this point, it is important to mention that the COVID 10 lockdown increased the difficulty of practicing PA due to mobility restrictions and fear of infection, particularly among older adults. This finding is supported by one study conducted by Hawkley et al., (2009) [40]. They suggested that those individuals who remain at home or are lonely are less active. Insufficient physical activity is also a risk factor for several non-communicable diseases. The lack of PA is defined as the fourth most common risk factor for mortality [41]. In contrast, regular physical activity can reduce the number of days of disease and also decrease the risk of upper respiratory tract infection [42]. Social isolation has been strongly associated with lower levels of PA. Therefore, isolated individuals are likely to report several types of health-related diseases [43]. PA is also effective in preventing work-related low back disorders. Back pain can be reduced by adopting a healthy lifestyle, which includes the regular practice of physical activity and ergonomic interventions [44]. 

Furthermore, significant differences were found between the intensity of the back pain reported by the subjects who complied with the ergonomic recommendations and those who did not (*p* < 0.001; ES = 0.17). Similarly, the respondents who reported suffering from mild or no stress reported significantly lower back pain as compared to those who suffered medium to high stress (*p* < 0.001; ES = 2.04). Moreover, the back pain intensity reported by the interviewees who were seated all or most of the time was significantly higher than those who were moving all or most of the time (*p* < 0.001; ES = 1.87). In this regard, it has been verified that quarantine duration is associated with worse mental wellbeing. Some studies analyzing the impacts of pandemics on mental health [45,46] have shown that stress disorders affect 28.9% of individuals, while depression affects 31.2%. This trend continued throughout the COVID-19 pandemic, as our respondents perceived the closure of schools as stressful. The respondents who reported mild or no stress (n = 645) presented a pain intensity of 2.83 (1.31). On the contrary, the interviewees who reported higher stress levels (n = 94) presented a back pain intensity of 3.66 (0.93). In connection with these results, Yue et al., (2012) indicated that women are more likely to suffer from emotional exhaustion. This might be the reason for their MSD higher prevalence as compared to male teachers [13]. 

Social isolation due to lockdown affects health behaviors through its impact on social support [47]. The quarantine has been linked to mental health deterioration in Slovak primary school teachers [48]. This author verified that the average stress level of experienced teachers was comparatively higher than the stress suffered by new teachers. However, both primary and secondary school teachers faced serious challenges at work during the COVID-19 lockdown. This is reflected in the relatively high percentage of respondents who indicated that they suffered from moderate or severe stress. Before the COVID-19 pandemic, Moccia et al. reported that 38% of the general population perceived some form of mental distress or anxiety. The COVID-19 pandemic could have exacerbated mental health conditions in patients with chronic pain, and negatively affect access to pain treatments [49]. These patients perceive a deterioration in their quality of life, with increased pain and depression [49].

Our article also has some limitations, but there are also significant strengths. As for the strengths, this study reflects the situation of Slovak pedagogues during quarantine. At the same time, direct relationships between MDS, PA habits, psychological aspects, ergonomics, and the occurrence of musculoskeletal pain during locomotion have been identified. As for the limitations, since the survey was conducted online, the control over the subjects who participated was limited, and there was no detailed medical information available about previous pain, treatments, and medication. Future studies should focus on the correlation between physical activity and chronic pain in teachers and assess the impact of online learning on work-related health burdens.

## 5. Conclusions

The highest back pain incidence in female teachers was reported in the cervical region, followed by the low back pain region. Over 60% of the respondents reported pain in more than one spinal region. The prevalence of back pain reported by secondary school teachers was significantly higher than the reported by primary school teachers. The most important risk factor for suffering from back pain was lack of PA (3.86). The highest prevalence of back pain was found in the region of Prešov. Based on the study results, it is necessary to provide teachers working in Slovakia with information about the possibilities and strategies to prevent MSD, emphasizing the motivation to change risky behaviors. PA could significantly reduce back pain intensity in female teachers, but further studies are needed to verify the real role of this factor. The data obtained regarding how home-based work affects teachers’ musculoskeletal health might be important for proposing future measures to reduce the burden of this work system. The main challenges are: insufficient PA, adopting sustained static or poor postures, and not complying with the ergonomic working recommendations conditions. Psychological aspects such as physical and mental stress while working at home can result in musculoskeletal pain and reduced productivity. The situation in the education sector should be monitored over time as the pandemic continues to impact teachers’ daily work.

## Figures and Tables

**Table 1 healthcare-09-00860-t001:** Spinal pain prevalence by vertebral segment.

Epidemiological Aspects	Percentage
Individuals suffering from cervical pain	74.84%
Individuals suffering from dorsal spine pain	29.12%
Individuals suffering from lower back pain	67.68%
Individuals who not suffering from lower back pain	5.36%
Individuals suffering from spinal pain in more than one area	61.31%

**Table 2 healthcare-09-00860-t002:** Pain intensity comparisons based on different conditions or factors.

Factor	Component/category	Pain Intensity ●
Regions *	Trnava (*n* = 62)	3.38 (1.14)
	Nitra (*n* = 94)	3.42 (0.94)
	Žilina (*n* = 141)	3.51 (1.09)
	Bratislava (*n* = 83)	3.52 (1.01)
	Banská Bystrica (*n* = 92)	3.53 (0.99)
	Trenčín (*n* = 66)	3.58 (0.88)
	Košice (*n* = 140)	3.61 (1.14)
	Prešov (*n* = 104)	3.74 (0.91)
BMI	Underweight (*n* = 24)	3.37 (1.01)
	Normal weight (*n* = 386)	3.52 (1.05)
	Overweight (*n* = 244)	3.61 (0.99)
	Obese (*n* = 128)	3.55 (1.05)
School type *	Primary school (*n* = 496)	3.49 (1.03)
	Secondary school (*n* = 225)	3.61 (1.01)
	Special education (*n* = 36)	3.83 (0.98)
	University (*n* = 25)	3.52 (1.08)
Number of days a week of online classes *	Once a week (*n* = 36)	3.33 (1.17)
	Twice a week (*n* = 23)	3.17 (1.07)
	Thrice a week (*n* = 50)	3.46 (1.18)
	Four times a week (*n* = 58)	3.51 (1.08)
	Five times a week (*n* = 615)	3.58 (1.01)
Years of teaching	1–4.9 (*n* = 128)	3.44 (1.03)
	5–9.9 (*n* = 88)	3.36 (1.07)
	10–14.9 (*n* = 116)	3.64 (1.02)
	15–19.9 (*n* = 122)	3.56 (1.08)
	20–29.9 (*n* = 205)	3.66 (0.99)
	+30 (*n* = 123)	3.55 (1.07)
Weekly teaching hours	1–9.9 (*n* = 194)	3.38 (1.16)
	10–19.9 (*n* = 401)	3.59 (0.97)
	20–29.9 (*n* = 165)	3.61 (0.99)
	30–39.9 (*n* = 20)	3.80 (1.06)
	40+ (*n* = 2)	3.5 (0.50)
Weekly practice of PA (days a week) *	None (*n* = 108)	3.86 (1.01)
	1–2 (*n* = 255)	3.63 (0.93)
	3–4 (*n* = 263)	3.52 (0.94)
	5–6 (*n* = 87)	3.15 (1.22)
	7 (*n* = 69)	3.37 (1.28)
Ergonomic recommendations	Subjects who complied with the ergonomic recommendations (*n* = 376)	3.55 (1.04)
Compliance #	Subjects who did not comply with the ergonomic recommendations (*n* = 99)	3.59 (0.98)
Stress #	Mild or no stress (*n* = 645)	2.83 (1.31)
	Moderate or severe (*n* = 94)	3.66 (0.93)
Time spent sitting or moving #	Subjects who were sitting all or most of the time (*n*= 613)	3.62 (0.97)
	Subjects who were moving all or most of the time (*n* = 70)	2.84 (1.29)

● pain was rated by the interviewees from 1 to 5. * Significant main effect observed. # Significant difference observed.

**Table 3 healthcare-09-00860-t003:** Correlations between spinal pain intensity and personal and environmental factors.

Factor	Pain Intensity	
Age	*r* = 0.014	*p* = 0.691
BMI	*r* = 0.066	*p* = 0.066
Years of teaching	*r* = 0.021	*p* = 0.55
Number of days a week having online classes	*r* = −0.05	*p* = 0.166
Weekly teaching hours	*r* = 0.034	*p* = 0.336
Time spent sitting or moving	*r* = 0.013	*p* = 0.715
Weekly frequency of PA *	*r* = −0.151	*p* < = 0.000
Ergonomic recommendations compliance *	*r* = - 0.109	*p* = 0.002
Perceived stress *	*r* = 0.150	*p* < 0.000

* significant correlation observed; *r*: Pearson correlation; *p*: significance level was set at <0.05.

## Data Availability

The data presented in this study are available on request from the corresponding author.

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
