# Peer review of "Back Pain Prevalence, Intensity, and Associated Risk Factors among Female Teachers in Slovakia during the COVID-19 Pandemic: A Cross-Sectional Study"

_healthcare, 2021, doi:10.3390/healthcare9070860_

Round 1

Reviewer 1 Report

A really interesting paper and relevant with the current climate with many school teachers working from home without the adequate equipment. 

Referencing mistake in the intro Wáng X., Wáng J., Káplár (2016) - normally you would not use first initials? 

Method section  - mistake should only be 26 not 2626  - Guidelines for Health-Related Research Involving Humans (2016), [2626]

Could you have used a reading scale example - Flesch Reading Scale 

Author Response

Vážený recenzent 1

Veľmi pekne ďakujem za vaše pripomienky. Všetky z nich boli označené a pomohli nám vylepšiť kvalitu článku. Úpravy, ktoré navrhol Recenzent 1, boli napísané červenými písmenami, Recenzent 2 modrými písmenami a Recenzent 4 hnedými písmenami. Keď dvaja recenzenti navrhujú rovnakú úpravu textu, podčiarkli sme text upravený konkrétnou farbou iného recenzenta.

Odkazy na chybu v úvode Wáng X., Wáng J., Káplár (2016) - obvykle by ste nepoužili prvé iniciály?

Bolo to upravené.

Časť Metóda - chyba by mala byť iba 26, nie 2626 - Pokyny pre výskum súvisiaci so zdravím ľudí (2016), [2626]

Bolo to upravené.

Mohli ste použiť príklad stupnice čítania - Fleschova stupnica čítania

Z našej skromnej perspektívy sme sa pri písaní tohto článku snažili používať zrozumiteľný, priamy a jednoduchý jazyk. Snažili sme sa zmeniť zložitý písaním a dlhým vetám. Rozumieme preto, že tento text je vhodný pre pokyny pre časopis.

Reviewer 2 Report

Dear Author, 

Thank you for the opportunity to review the paper entitled “Back pain prevalence, intensity, and associated risk factors among female teachers in Slovakia during COVID-19 pandemic: A Cross-Sectional Study”. The paper deals with a very interesting topic. The selection of methodology and statistical analysis are to be commended. Good scientific sound, but the paper still needs to be revised, mostly editorially. Below is a list of my comments:

Major:

  • The objectives of the paper diverge between the abstract, introduction and discussion. It must be the same (Unfortunately I can't point to the lines because the manuscript doesn't contain them)

Minor issues:

  • Inconsistent recording of percentages: "Seventy-four per cent reported significant cervical pain" or "67% low back pain"
  • I suggest indicating the number of participants by numbers in the abstract
  • The abstract did not define the abbreviation PA, I guess it is about physical activity?
  • However, in the introduction, the abbreviation PA is translated twice
  • The readability of the 3rd paragraph of the introduction is quite limited. Themes of physical activity, COVID restrictions, sedentary lifestyle are mixed up - please work on making this part clearer
  • The objectives of the paper diverge between the abstract and the introduction. It must be the same (Unfortunately I can't point to the lines because the manuscript doesn't contain them)
  • The reference to "World Health Organization a Council for International Organizations of Medical Sciences - 2017 - International Ethical Guidelines for Health-Relate.Pdf. Dostępne online: https://cioms.ch/wp-content/uploads/2017/01/WEB-CIOMS-EthicalGuidelines.pdf" is inserted twice in the text.
  • The sentence "The purpose of the survey, its introduction, and the length have been completed within the open electronic online survey framework" sounds strange, please change the sentence style
  • This sentence doesn't sound right either, the criterion is ambiguous: "The exclusion criteria were: i) Being a male"
  • Information on survey distribution channels duplicated in manuscript

Author Response

Dear Reviewer 2

Thank you very much for your comments. All of them were taken into account and helped us to improve the quality of the article. Modifications suggested by Reviewer 1 have been written in red letters, by Reviewer 2 in blue letters, and by Reviewer 4 in brown letters. When two reviewers suggested the same text modification, we underlined the text modified with a particular color of another reviewer.

The objectives of the paper diverge between the abstract, introduction and discussion. It must be the same (Unfortunately I can't point to the lines because the manuscript doesn't contain them).

The wording of the study objective in the three mentioned sections (abstract, introduction and discussion) has been modified to guarantee coherence.

Inconsistent recording of percentages: "Seventy-four per cent reported significant cervical pain" or "67% low back pain"

This aspect has been corrected in the abstract to report the data consistently.

I suggest indicating the number of participants by numbers in the abstract.

It has been modified.

The abstract did not define the abbreviation PA, I guess it is about physical activity?

However, in the introduction, the abbreviation PA is translated twice

It has been corrected, both in the abstract and introduction.

The readability of the 3rd paragraph of the introduction is quite limited. Themes of physical activity, COVID restrictions, sedentary lifestyle are mixed up - please work on making this part clearer.

The 3rd paragraph has been modified and drafted clearly. 

The reference to "World Health Organization a Council for International Organizations of Medical Sciences - 2017 - International Ethical Guidelines for Health-Relate.Pdf. Dostępne online: https://cioms.ch/wp-content/uploads/2017/01/WEB-CIOMS-EthicalGuidelines.pdf" is inserted twice in the text.

It has been modified. One citation of 26 was deleted.

The sentence "The purpose of the survey, its introduction, and the length have been completed within the open electronic online survey framework" sounds strange, please change the sentence style.

It has been modified.

This sentence doesn't sound right either, the criterion is ambiguous: "The exclusion criteria were: i) Being a male"

It has been modified.

Information on survey distribution channels duplicated in manuscript.

The duplicate sentence has been removed.

Reviewer 3 Report

The interest to the readers is limited, the article does not provide evidence for clinical practice

Author Response

Dear Reviewer 3,

Thank you very much for your comments. All of them were taken into account and helped us to improve the quality of the article. Modifications suggested by Reviewer 1 have been written in red letters, by Reviewer 2 in blue letters, and by Reviewer 4 in brown letters. When two reviewers suggested the same text modification, we underlined the text modified with a particular color of another reviewer.

The interest to the readers is limited, the article does not provide evidence for clinical practice

According to our humble opinion, the study provides valuable information for a wide range of readers. Currently, there is no similar studies monitoring the teachers' health status during the pandemic. The epidemiological and health-related information provided is crucial to plan the future education mode during the following months or even years. Additionally, several adjustments were made to increase the quality of the article presented.

Reviewer 4 Report

I think that the study is quite interesting and well conducted and there are not a lot point which need to be further addressed except the limitations.

Abstract: define PA

Introduction:

The introduction is well written, inromative and guides to the study purpose.

  • Please end the introduction with a hypothesis.
  • Please write one sentence that the data of your study can be relevant not only during the pandemic situation, but also for the future concerning the growing sector of remote learning.

Methods

  • subjects: why you have not used e-mail to contact schools and establish direct contact. How could you control that persons with different profession participate, because facebook groups of teachers might be open for everybody. Please define this as a limitation.
  • Did you ask if there have already been spine symptoms before starting remote learning?

Results

  • nearly half of the study population has overweight/obese. Is this representive fort he teachers in your country or does this represent a selection bias.

Discussion

  • first line of discussion. Please be precise. You were not examining musculoslecetal disorders in general, the questionnaires were specific for spind disorders.
  • Please define further limitations as mentioned above.

Author Response

Dear Reviewer 4,

Thank you very much for your comments. All of them have been taken into account and will be helpful to improve the quality of the article. Hereunder we detail the modifications made in the article. Modifications suggested by Reviewer 1 have been written in red letters, by Reviewer 2 in blue letters and by Reviewer 4 in brown letters. When two reviewers suggested the same text modification, we underlined underlined the text modified with a particular color of another reviewer.

Abstract: define PA

It was modified

Introduction:

Please end the introduction with a hypothesis.

The hypothesis was moved to the end of the introduction.

Please write one sentence that the data of your study can be relevant not only during the pandemic situation, but also for the future concerning the growing sector of remote learning.

The sentence was added, thank you.

Methods

subjects: why you have not used e-mail to contact schools and establish direct contact. How could you control that persons with different profession participate, because facebook groups of teachers might be open for everybody. Please define this as a limitation.

We disseminated the questionnaire exclusively in Facebook groups whose access is limited to teachers who possess the International Teacher Identity Number in Slovakia. Even so, we added this as a limitation of the study at the end of the discussion. Thank you.

Did you ask if there have already been spine symptoms before starting remote learning?

We have focused on the point prevalence of one particular period. We did not consider it relevant based on our modest opinion since the pain recalls from the time ¨before the remote learning¨ is a long period of time (more than one year). Furthermore, according to some authors, pain recall is not reliable or accurate  (Daoust, R., Sirois, M. J., Lee, J. S., Perry, J. J., Griffith, L. E., Worster, A., Lang, E., Paquet, J., Chauny, J. M., & Émond, M. (2017). Painful Memories: Reliability of Pain Intensity Recall at 3 Months in Senior Patients. Pain research & management, 2017, 5983721. https://doi.org/10.1155/2017/5983721)

Results

nearly half of the study population has overweight/obese. Is this representive for the teachers in your country or does this represent a selection bias.

Yes, we consider that it represents the slovakian female teachers for the following reasons: i) there was no limitation for teacher selection (apart from sex). ii) The sample size is very large. iii) There are respondents from all regions of Slovakia. However, there is no similar studies in Slovakia to set comparisons. Even so, the overweight prevalence is very similar to the other studies from different parts of the world:

Brazil - 47.2% (Rocha et al.)

https://periodicos.ufsc.br/index.php/rbcdh/article/view/1980-0037.2015v17n4p450/29628

India - 43.2% Grade I obesity, 20.4% Grade II obesity and 6.6% had Grade III obesity (Sarah Jane Monica, Sheila John, Madhanagopal. R)

DOI : http://dx.doi.org/10.12944/CRNFSJ.6.2.15

Ghana 34% (Pobee R., Owusu W., WA Plahar)

https://www.ajol.info/index.php/ajfand/article/view/90623/80038

Discussion

first line of discussion. Please be precise. You were not examining musculoslecetal disorders in general, the questionnaires were specific for spind disorders.

This sentence was adjusted to be more specific and aligned with the research objective.

Please define further limitations as mentioned above.

The above-mentioned limitations have been added.

Round 2

Reviewer 3 Report

I am agree with the autor's response. 

Reviewer 4 Report

I have no further points which need to be addressed.